# Effect of Brewers’ Spent Grain Addition to a Fermented Form on Dough Rheological Properties from Different Triticale Flour Cultivars

**DOI:** 10.3390/foods14010041

**Published:** 2024-12-27

**Authors:** Aliona Ghendov-Mosanu, Sorina Ropciuc, Adriana Dabija, Olesea Saitan, Olga Boestean, Sergiu Paiu, Iurie Rumeus, Svetlana Leatamborg, Galina Lupascu, Georgiana Gabriela Codină

**Affiliations:** 1Faculty of Food Technology, Technical University of Moldova, 9/9 Studentilor St., MD-2045 Chisinau, Moldova; aliona.mosanu@tpa.utm.md (A.G.-M.); olga.boestean@tpa.utm.md (O.B.); sergiu.paiu@doctorat.utm.md (S.P.); rumeus.iurie@usch.md (I.R.); 2Faculty of Food Engineering, “Stefan cel Mare” University, 720229 Suceava, Romaniaadriana.dabija@fia.usv.ro (A.D.); 3Faculty of Economics, Engineering and Applied Sciences, Cahul State University “Bogdan Petriceicu Hasdeu”, MD-3909 Cahul, Moldova; 4Applied Genetics Laboratory, Institute of Genetics, Physiology and Plant Protection, Moldova State University, 20 Padurii St., MD-2002 Chisinau, Moldova; svetlana.leatamborg@sti.usm.md (S.L.); galina.lupascu@sti.usm.md (G.L.)

**Keywords:** brewers’ spent grain, fermentation process, dough rheological properties, triticale flour, principal component analysis

## Abstract

Triticale grains and brewers’ spent grain (BSG) can be new sources to develop food products. From a socio-economical point of view, this fact is important since triticale is easily adapted to the climatic changes and BSG is a low-cost material which may lead to a “zero-waste” desiderate. In this study, dough rheological properties obtained from different triticale cultivars (Ingen 33, Ingen 35, Ingen 54, and Ingen 93) cultivated in the Republic of Moldova and BSG in a fermented form (BSF) in an addition level of 10% and 17.5% were analyzed. For this purpose, different rheological devices, such as Mixolab, Alveograph, HAAKE MARS 40 Rheometer, Falling Number, and Rheofermentometer, were used. Also, the pH value of the dough samples with different levels of BSF addition during fermentation was determined. According to the data obtained, BSF addition decreased water absorption values; torques values corresponding to stages 1–5 of the Mixolab curve; and dynamic rheological elastic, viscous, and complex modules. For the 17.5% BSF addition to triticale flour, the best rheological results were obtained for the Ingen 33 and Ingen 54 varieties. In addition, the BSF addition decreased the baking strength and tenacity of the Alveograph curve. The pH values of the dough samples during fermentation significantly decreased (*p* < 0.05) with the increased amount of BSF incorporated into the dough recipe. The highest pH decreased values were obtained for Ingen 35 with a 17.5% BSF addition, which varied between 5.58 and 5.48. During fermentation, all data recorded by the Rheofermentometer device were improved. The dough samples presented a high retention coefficient, which varied between 99.1 and 99.5%. The falling number decreased with the increasing level of BSF in triticale flour, indicating an increase in α-amylase activity in the mixed flours. The principal component analysis data showed a strong association between triticale flour varieties without a BSF addition and those with a high amount of BSF incorporated into the dough recipe. The results obtained indicate the fact that many mixes between BSF and different triticale varieties may lead to bakery products of a good quality.

## 1. Introduction

Triticale, a hybrid between wheat (*Triticum aestivum*) and rye (*Secale cereale*), combines the qualities of the two cereals: the resistance to harsh environments of rye and the nutritional value of wheat. Global triticale production now exceeds 20 million tons annually, largely concentrated in Europe, which generates over 80% of the world’s quantity [1,2]. Although there is significant variability, the nutritional composition of whole-grain triticale flour is closer to that of wheat than to that of rye. On average, triticale flour is characterized by a relatively high content of starch (60.8% to 67.6%), protein (11.8% to 15.2%), and dietary fiber (11.7% to 13.6%) [2]. The protein content of triticale is higher than that of rye and comparable to wheat, with a higher proportion of essential amino acids, such as lysine. Although triticale has a lower gluten content and quality, it compensates with its soluble protein; dietary fiber; and minerals, like phosphorus, potassium, magnesium, calcium, manganese, and iron. These nutritional characteristics make triticale an attractive option for baking, but the technological quality of its flour, characterized by high viscosity and low dough stability, limits its widespread use. In mixtures with wheat, rye, or other cereal flours, triticale flour can give better quality products with a high nutritional value [3,4,5].

The increased enzyme activity in triticale, particularly α-amylase, allows an improved digestibility of starch. However, this activity, combined with a high content of pentosans, can negatively influence baking properties. Blending triticale flour with other types of flour can balance these shortcomings, providing rheological stability and improved bakery products. Amylases, especially α-amylase ones, which may be present in low amounts in wheat flour, provide fermentable sugars for yeast cells and favor the production of carbon dioxide, which improves bread quality by increasing it loaf volume, crumb firmness, etc. [6,7]. According to Seguchi et al. [8], triticale flours of different varieties mixed with wheat flours in an addition level of 18.3% lead to bread samples with an increased height and specific volume. The use of triticale up to 50% in wheat flour of a strong quality for breadmaking led to bread samples of a high quality. Other products of a high quality that may be obtained from triticale flours are oriental noodles; high-fiber extruded snacks; and soft wheat-type products, such as layer cakes [9].

Beyond its use in food, triticale is also utilized in producing malt, beer, and animal feed. Products such as biscuits, tortillas, pasta, or bread can be formulated using triticale flour, either alone or in combination with other flours [10,11]. Recent research has demonstrated triticale’s potential as a functional ingredient due to its rich composition rich in arabinoxylans, *β*-glucans, and lunasin peptides, with beneficial effects on digestive health and prebiotic properties [12]. Triticale has shown adaptability in diverse food applications, but challenges remain in its processing. To increase consumer acceptability, it is necessary to develop new triticale varieties and processing methods that improve the technical properties of flour. In addition, greater promotion of the nutritional benefits of triticale could encourage wider use, especially in the context of the trend towards healthy and functional foods [13].

Triticale is a valuable cereal resource, combining a superior nutritional composition with varied possibilities for food and industrial use. However, to reach its full potential, the limitations of flour’s technological properties must be addressed and its health benefits promoted. Given the growing demand for food resources and interest in new products, triticale can become a sustainable and innovative solution in the food industry.

Although triticale flour is rarely used alone in baking, mixtures with wheat flour can compensate for these limitations, allowing technologically and sensorially acceptable baked products to be obtained.

The paper proposes the use of triticale flour with brewer’s spent grain (BSG), in its fermented form (BSF), to obtain bakery products. BSG is the main by-product of beer production, accounting for 85% of the total, with a global annual production of 39 million tons. It is rich in dietary fiber (41–59%); protein (18–24%), including essential amino acids; and phenolic compounds with antioxidant, anti-inflammatory, and anticancer properties. BSG has a high moisture content (70%), favoring the development of microorganisms and contributing to environmental pollution if disposed of improperly [14,15]. Each ton of discarded BSG generates the equivalent of 513 kg of CO_2_. Its nutritional composition varies with production and storage factors, including hemicellulose, cellulose, lignin, protein, lipids, and minerals such as silicon, phosphorus, and calcium. It also contains vitamins (B_1_, B_2_, B_6_, and K) and essential fatty acids, like linoleic acid. The amino acid lysine, predominant in BSG, makes it particularly valuable compared to other grains [16,17].

Although it has valuable nutritional content, the use of BSG in food is limited due to its effect on color, texture, and sensory acceptability, suggesting the need for further processing, such as lactic fermentation [17]. Through fermentation, BSG improves its bioactive profile, developing compounds such as flavonoids and phenolic acids, making it suitable for use in functional foods and animal feed [18]. Lactic fermentation of BSG facilitates the degradation of lignin, releasing blocked nutrients and favoring the regulation of the intestinal microbiota through the growth of beneficial bacteria, such as *Lactobacillus* and *Bifidobacterium*. These processes can contribute to the prevention of gastrointestinal diseases, diabetes, coronary heart disease, and cancer, supporting the circular economy by reinserting BSG into new food and bioactive production chains [19].

The use of BSG in fermented form as a sourdough in bread recipes may be a solution for its valorization which may help to obtain bakery products of an improved quality. The process involves the fermentation of a mixture of BSG and water by its own microorganisms, such as lactic acid bacteria and yeasts, which generate bioactive compounds such as amino acids, peptides, and antioxidants with potential beneficial health effects, including anticancer and antihypertensive properties [16]. Sourdough fermentation reduces anti-nutritional factors and dietary short-chain carbohydrates, promoting digestive health and diversifying gut microbiota. The study conducted by Galoburda et al. [20] found that the volatile component profile of triticale bread made with *Lactobacillus sanfranciscensis* sourdough and wheat mix dough without sourdough differed significantly. Compared to triticale bread without sourdough, the bread’s flavor was more perceptible due to the use of two-step sourdough, which allowed for the acquisition of a wider and more varied spectrum of volatile compounds. The resulting products have an improved flavor, are easier to digest, and provide an increased supply of essential nutrients [21,22]. From a technological point of view, sourdough fermentation extends the shelf life of bread through antimicrobial compounds produced by lactic acid bacteria, which inhibit mold growth and prolong freshness. The process stimulates the release of phenolic compounds and fatty acids, improving the flavor and nutritional value of the bread. It also has a positive impact on dietary fibers, adjusting their characteristics according to the degree of fermentation [23].

From a nutritional perspective, sourdough bread offers a low glycemic index, helps increase satiety, and can relieve gastrointestinal upset. However, the beneficial health effects are variable, requiring standardization of studies to demonstrate significant clinical impact. However, sourdough is recognized for its ability to improve mineral accessibility, especially in high-fiber products [24]. Recent research has highlighted its potential to create bakery products with distinct flavors and support sustainability trends by reducing food waste. In other words, sourdough represents a traditional technology with modern applications, combining innovation with cultural heritage [25]. It is a valuable tool to produce healthy and sustainable bread, responding to the demands of contemporary consumers for nutritious, tasty, and diversified products [26]. According to our knowledge, this is the first study which highlight the impact of brewer’s spent grain in a fermented form on triticale dough’s rheological properties to improve bakery products’ quality. For this purpose, two additional levels of BSF, namely 10 and 17.5%, were added to triticale flour in order to determine its impact on dough rheological behavior. Through this BSF addition level in triticale flour, we want to have an impact on bakery, pasta, or other products from the nutritional point of view. BSG represents an important source of protein, dietary fiber, and high content of bioactive substances (tannins, flavonoids, and phenolic acids), mineral substances, and vitamins with multiple benefits on the human body [27]. Through fermentation, the utilization of nutrients from BSG increases. At the same time, the fermentation process helps us to reduce the content of antinutritional factors in BSG, which can also have a positive effect on the sensory profile of the bakery, pasta, or other products obtained. However, a higher amount of BSF addition may lead to a decrease in the quality of the final products, especially due to the gluten dilution effect from the triticale flour. The main objective of this study was to analyze the influence of the BSF addition to triticale flour on the dough’s rheological properties according to the following measures: variation in the value of the falling number; variation in the dough rheological properties during mixing, heating, and extension; variation in the flour’s ability to form and retain the gases formed during fermentation process; and the variation in the dough’s fundamental rheological properties. Also, the use of four triticale varieties helped us to obtain proper conclusions regarding BSF’s impact on their rheological behavior during the technological process of bakery, pasta, or other products.

## 2. Materials and Methods

### 2.1. Triticale Flour Samples

Four triticale grains varieties cultivated in the Republic of Moldova were used in this study. These varieties were Ingen 33, Ingen 35, Ingen 54 and Ingen 93 which were received from the Institute of Genetics, Physiology and Plant Protection, the Republic of Moldova. The triticale varieties were from the 2023 harvest. To obtain flour, triticale grains were ground in a laboratory mill 3100 (Perten Instruments, Hägersten, Sweden) and analyzed for their protein, ash and wet gluten content according to the ICC 105/2, ICC 104/1 and ICC 137/1 methods [28]. The triticale flours presented the following data: 13.19–15.1% for protein content, 1.42–1.49% for ash content, and 19.01–27.21% for wet gluten content obtained from triticale flour, with a minimum value for Ingen 94 and a maximum one for Ingen 33.

### 2.2. Brewers’ Spent Grain Sourdough Fermentation

The brewers’ spent grain (BSG) resulted from the blonde beer process, were received from the Î.M. “Efes Vitanta Moldova Brewery” S.A., Chisinau, the Republic of Moldova and were dried at 40 ± 1 °C up to a moisture value of 6.3 ± 0.1% according to the ICC 110/1 method [27]. After that the BSG was milled through a mill feeder (Model 3100, Perten Instruments, Segeltorp, Sweden) and stored at room temperature at 20 °C in polyethylene bags until it uses for sourdough fermentation. The brewers’ spent grain flour (75 g) was mixed with water (200 mL) and 25 g triticale flour to create a liquid dough. It was placed in a proofing chamber at 30 ± 1 °C and 85% relative humidity, for 24 h. Every 24 h the dough was refreshed by adding another mixture of nutrient medium (100 g of fermented dough, 200 g flour and 200 mL water) until the dough reached an acidity of 8–10 degrees and a pH of 3.78 ± 0.01. The pH and total titratable acidity value (TTA) value of the fermented brewers’ spent grain (BSF) were determined according to the standard methods 90:2007 [29]. The TTA value was defined as the amount of 0.1 N NaOH solution (mL) used to neutralize 10 g sample weight. The pH was determined using an HQ30d portable pH Meter (HACK, Loveland, CO, USA). The BSF was incorporated into the triticale dough at levels of 10 and 17.5%.

### 2.3. Dough Rheological Properties During Mixing and Pasting

Mixolab device (KPM, Tripette et Renaud, Paris, France) was used to evaluate rheological properties during mixing and pasting of the dough obtained from triticale flour in which different amounts of brewer’s spent grain sourdough fermentation (BSF) were incorporated according to the ICC No. 173 standard method [28]. The following dough rheological properties during mixing were determined: water absorption (WA), which shows the water amount that dough needs to reach the optimum dough consistency, corresponding to C1 torque (1.1 Nm); dough development time (DDT); and dough stability (ST). Dough rheological properties recorded by Mixolab during heating were torques C2, C3, C4, and C5, which correspond to protein weakening, starch gelatinization, stability of hot starch paste, and final starch paste viscosity after cooling at 50 °C. The differences between Mixolab torques C1 and C2 (C12), C3 and C2 (C32), C3 and C4 (C34), and C5 and C4 (C54) offer information about protein weakening, starch gelatinization, amylolytic activity, and starch gelling.

### 2.4. Dough Dynamic Rheological Properties

Fundamental dough rheological properties were determined using a HAAKE MARS 40 rheometer (Thermo-HAAKE, Karlsruhe, Germany) with parallel plate geometry of 40 mm in diameter. Between plates, the dough samples were placed after a rest for 10 min to allow for relaxation. Dough excess was removed, and a gap of 2 mm was used. To avoid dough drying during analysis, a thin layer of Vaseline was applied. Frequency sweep tests from 1 to 20 Hz were carried out in the linear viscoelastic region (LVR) previously established. The storage modulus (G’), loss modulus (G”), and loss tangent (tan δ) were determined at a constant stress of 15 Pa.

### 2.5. Dough Rheological Properties During Extension

An Alveograph (KPM, Tripette et Renaud, Paris, France) was used to determine the dough rheological properties during extension, according to ICC 121, at a 14% moisture basis at constant hydration. The following parameters were determined: maximum pressure (P), index of swelling (G), dough extensibility (L), configuration ratio of the Alveograph curve (P/L), and baking strength (W).

### 2.6. Dough Rheological Properties During Fermentation and Falling Number Values

A Rheofermentometer device (Chopin Rheo, type F4, KPM, Tripette et Renaud, Paris, France) was used to determine dough rheological properties during fermentation as follows: total CO_2_ volume production (VT, mL), volume of the gas retained in the dough at the end of the test (VR, mL), maximum height of gaseous production (H’m, mm), and retention coefficient (CR, %). For this analysis, dough samples were obtained by mixing 250 g triticale flours and BSF addition, 5 g salt, and 3 g dry yeast of the *Saccharomyces cerevisiae* type. The falling number value (FN, s) was determined with a Falling Number device (FN 1305, Perten Instruments AB, Stockholm, Sweden).

### 2.7. Statistical Analysis

Analysis of variance (ANOVA) was used to analyze the data with Tukey’s test at a significance level of α = 0.05. For this purpose, Statgraphics software Centurion XVI 16.1.17 (Statgraphics Technologies, Inc., The Plains, VA, USA) was used. For principal component analysis, the statistical program XLSTAT 2021.2.1 software (Addinsoft, New York, NY, USA) was applied.

## 3. Results and Discussion

### 3.1. Dough Rheological Properties During Mixing and Pasting

The Mixolab data obtained for triticale dough samples with different levels of BSF addition are shown in Table 1. As may be seen the BSF addition significantly (*p* < 0.05) decreased water absorption level which indicates that triticale flour needs less water to reach the optimum consistency, which may be an economic disadvantage for breadmaking producers. However, this parameter does not have a direct impact on bakery products’ quality. It only indicates how much water the processors must add in order to obtain the optimum dough consistency. Dough stability time (ST) time decreased with the increased level of BSF addition to triticale flour, whereas the dough development time (DDT) presented some fluctuations. These results are in disagreement with those reported by Aprodu et al. [30] when BSG was incorporated into wheat flour. However, when sourdough was used in dough recipe, dough stability decreased with the increase level of BSF addition to triticale flour. The variation in this parameter depends on the triticale variety. With a 10% BSF addition level in triticale flour, no significant (*p* < 0.05) variation was obtained for Ingen 33, Ingen 35, and Ingen 93 varieties. Moreover, for Ingen 54, a slight improvement in this parameter value may be noticed when the highest amount of BSF was incorporated into the dough recipe. Dough development time (DDT) values fluctuated for dough samples with different amounts of BSF added to triticale flour. Generally, the value of this parameter increased when 10% of BSF was incorporated into triticale flour; however, it decreased in the Ingen 33 variety when 17.5% BSF was used in dough recipe. During heating, dough behavior is also influenced by the triticale variety and concentration of the BSF addition. The highest values of these parameters were obtained for the Ingen 54 variety, probably due to its lower proteolytic activity, as may be seen from Mixolab data values C2 and C12. Also, the BSF addition changed gluten quality due to a wide range of metabolites produced during fermentation that affect protein composition, including gluten, which plays an essential role in dough quality [31]. The triticale proteins’ hydrolysis due to BSG sourdough fermentation weakens the strength of triticale dough and significantly (*p* < 0.05) decreases ST and DDT values. The main changes in dough rheological properties are due to the impact of the pH decreased values, probably due to the effect that the BSF addition to triticale flour has on the dough components. The structural changes in the dough are largely due to the action of the enzymes in the flour, as they approach or reach their optimum activity. This favors protein denaturation, especially as the temperature increases and reaches the optimal activity temperature of proteolytic enzymes [32,33]. Their activity significantly increased for dough prepared with sourdoughs, a fact highlighted by the significant (*p* < 0.05) increase in C12 Mixolab values with the increasing amount of BSF addition to dough samples for all triticale varieties.

Also, an important technological role in the triticale breadmaking process is influenced by arabinoxylans, compounds with a high water-binding capacity and a key role in the viscosity of triticale dough [32]. The viscosity of the triticale dough decreased during the heating process for the dough with the BSF addition due to the intensification of the enzyme degradation action of the arabinoxylans [34]. This, along with protein denaturation process, will significantly (*p* < 0.05) decrease the C2 Mixolab value for all dough samples with a BSF addition to triticale flour. During heating, the gelatinization process of the starch from the dough system takes place. This process is a complicated one due to the complexity of the dough system, the lack of water in the dough, and the presence of compounds that bind water and compete with starch for this water [35]. Since the amount of water is relatively low, not all starch granules will gelatinize; some will only swell without destroying the granular structure. Along with these swollen granules, there will be gelatinized granules that have not been fractionated, and so they have kept their form; gelatinized and fractionated granules; dispersed granules, in which the content was released and formed an actual gel; and enzymatically degraded starch granules [36]. An important aspect is related to the enzymatic starch hydrolysis, which can absorb a higher amount of water from the dough system and can gelatinize completely [37]. The Mixolab parameters related to the starch gelatinization process are C3 and the difference between torques C3 and C2 (C32). As may be seen, these values significantly decreased (*p* < 0.05) with the increasing amount of BSF addition, showing that the gelatinization capacity of the dough samples decreased. This may be due to lower starch content from the triticale–BSF composite flour samples. As the amount of BSF in triticale flour increased, the starch compound from the dough system decreased, and when the temperature increased above the starch gelatinization value, this led to a decrease in the dough viscosity. Also, the C3 value will decrease with the decrease in the starch gelatinization capacity and the increase in the *α*-amylase activity [38]. The *α*-amylase activity is influenced by pH and temperature. The BSF addition to triticale dough led to a more acidic dough, with a pH value around 5.7 and 5.6 for dough samples with 10 and 17.5%. BSF additions to triticale flour are more related to the optimum level of the amylase’s enzyme activity, so they will increase their activity fast, as reflected by a decrease in the difference between torques C3 and C4 (C34) for dough samples with a BSF addition. *α*-Amylase experiences heating resistance around 80 °C and is thermally inactivated at 90 °C, meaning that it will act in the dough for quite a long time on an easily accessible substrate, even if at a lower speed [39]. The result of this action is a more intense starch hydrolysis with accumulation of dextrins, which will determine the reduction in the amount of gelatinized starch [38]. This will reduce dough consistency fact reflected by a decrease in C4 value, the Mixolab torque for stability which indicates hot starch paste. Similar data were reported by Neylon et al. [40], who noticed, while using a Rapid Visco Analyzer, a reduction in peak and final viscosities for semolina flour with BSF addition. The starch retrogradation and difference between torques C5 and C4 (C54) presented some fluctuations related to the BSF addition to triticale dough. An enzymatically attacked starch will retrograde less, but BSF addition to triticale flour will change the three-dimensional network of dough due to its composition, including non-starch polysaccharides, non-gluten proteins, etc., which may lead to a low retrogradation process of bakery products.

### 3.2. Dynamic Rheological Properties of Dough Samples

The BSF addition to triticale flour led to a significant (*p* < 0.05) decrease in all dynamic rheological values for dough samples: elastic modulus (G’), loss modulus (G”), and complex modulus (G*). Regarding the phase angle value (tan δ) value, it significantly increased by incorporating BSF into triticale flour, indicating that the dough becomes more viscous.

According to the data presented in Figure 1 and Table 2 in which modules G’, G”, and G* and tan δ data are shown at 10 Hz, it may be seen that the BSF incorporation into the dough recipe led to a decrease in G’, G”, and G* and an increase in tan δ values, showing a decrease in dough consistency. For whole frequency range, the elastic modulus (G’) values a higher than the viscous ones (G”), indicating a solid elastic-like behavior for all dough samples [40]. The tan δ is lower than 1 for all the dough samples, showing that the viscous part of the dough is less prominent than the elastic one [41]. Compared with the samples with a BSF addition, the control one presented the lowest tan δ values. This indicates the fact that BSF incorporation in triticale flour leads to a loss of elasticity in dough samples. The analysis performed for dough samples in which BSF was incorporated into triticale flour showed an initial decrease in the phase angle with frequency, followed by an increase in their values. 

The increase in the phase angle value and the decrease in the G’ values due to the incorporation of BSF in triticale flour indicate a dough less elastic, but at the same time less firm. Elasticity is provided by gluten, but especially by glutenin, and consists in the fact that the dough is deformed reversibly up to a certain applied force [42]. The BSF addition to triticale flour reduces the gluten content of the dough, including the glutenin content, thus leading to a significant decrease in the G’ values. Also, the decrease in G” values with the increase in the amount of BSF added to triticale dough is a consequence of the decrease in gluten content in the dough. The BSF addition may increase the soluble fraction from the dough system. This fact is reflected in lower water absorption values and dough development times, a fact highlighted by the Mixolab device data. Increasing the content of water-soluble substances reduces G’ and G” values and increases tan δ value, showing that they make the dough more viscous [43]. Higher G’, G”, and G* values and lower tan δ values indicate that the dough is of a good quality for breadmaking because it is more elastic. According to our data, the BSF addition to triticale flour decreased the quality of dough for breadmaking. As the level of BSF addition to the dough recipe increased, the dough enzymatic activity was higher, and thus, the degradation of starch and proteins was intensified, which led to a decrease in viscoelastic modules. A similar trend was also reported by Zhang et al. [44], who underlined the impact of maltohexaose-producing *α*-amylase on rheological behavior of dough.

### 3.3. Dough Rheological Properties During Extension

According to the Alveograph data shown in Table 3, it may be seen that the BSF addition to triticale flour had a significant (*p* < 0.05) influence on dough tenacity, baking strength, and the configuration ratio, according to the Alveograph curve. Also, it presented a significant effect (*p* < 0.05) on dough extensibility and index of swelling, especially when high levels of BSF were incorporated into the dough recipe.

A decrease in dough tenacity (P) and baking strength (W) may be due to the action of enzymatic activity on gluten and starch, which weaken the dough. Also, BSF does not contain gluten, which is formed during mixing, leading to dough with less resistance. A decrease in dough tenacity and baking strength may be related to the Alveograph standard method of analysis. According to it, dough rheological properties are determined at constant hydration, with the same amount of water addition. Considering Mixolab data, dough with BSF addition needs less water to reach the optimum consistency value. This means that dough will have a higher amount of water than it needs, thus causing it to soften and have lower values for P and W. For Ingen 33 and Ingen 54 varieties, the addition of 10% BSF in triticale flour led to a slight increase in extensibility and index of swelling values. This behavior may be because amylases and proteases act on starch and gluten. However, generally, these Alveograph parameters decreased due to the acids formed because of BSG fermentation. They reduce the pH in the dough, which causes a reduction in its extensibility by moving the pH toward to the isoelectric pH values of some protein fractions which precipitate and participate in the gluten structure, in agreeance with that reported by Codină et al. [45] when dry sourdough was incorporated into wheat flour. Also, the gluten content decreased due to the BSF addition to triticale flour, a fact that reduces dough extensibility. The ratio P/L significantly (*p* < 0.05) decreased with the increased amount of BSF added to triticale flour. However, except for the Ingen 35 variety, the dough samples with the same amount of BSF addition did not present significant changes regardless of the variety of triticale used. The decrease in the P/L ratio value was mainly due to the weak interactions that may take place in the dough system between the chemical compounds from BSF and triticale flour.

### 3.4. Effect of Brewer’s Spent Grain Sourdough Fermentation on pH, Dough Rheological Properties During Fermentation, and Falling Number Values

The BSF addition to triticale flour led to a significant decrease (*p* < 0.05) in the pH value of dough samples, as may be seen from Table 4. The decreased value of pH when different amounts of sourdough are incorporated into dough recipes was previously reported by different researchers [46]. This behavior is due to the lactic acid content of BSF, which is a result of the lactic acid bacteria fermentation in triticale flour. Similar data were also reported by Codină et al. [45] when dry sourdough was incorporated into wheat flour.

By using BSF in triticale dough, all parameters’ values obtained to Rheofermentometer device were improved, as can be seen from Table 5.

By introducing BSF into the dough, which contains lactic acid and bacteria in large numbers, it ensures that lactic fermentation begins immediately. The acidity introduced with BSF ensures a higher initial acidity in the dough and consequently ensures a high speed of alcoholic and lactic fermentation. When dough is prepared with BSF, a higher initial level of acidity is obtained, which ensures the existence of a large number of active lactic acid bacteria and rigorous yeasts, adapted to metabolizing maltose. This intense fermentation ensures gas formation at a good rate, a fact that may be seen from higher values of VT and H’m. Also, the addition of BSF to triticale flour increases the *α*-amylase activity, which already presents higher values according to the FN values [38]. This increase will favor maltose formation, which will be used by yeast to produce carbon dioxide and other fermentation products. Additionally, the increased amount of lactic acid bacteria due to the BSF addition to triticale dough will change its structural conformation and its ability to retain carbon dioxide formed. As may be seen the VR values significantly increased (*p* < 0.05) with the increased level of BSF added to triticale flour. The increase in the carbon dioxide formed during fermentation and dough ability to retain gases for dough samples with BSF addition led to a significant (*p* < 0.05) increase in H’m and CR values [47]. The CR value is the ratio between the Rheofermentometer parameters VR and VT. High values of this parameter indicate that dough’s ability to retain gases is higher than the amount of gases formed during the fermentation process. [48]. It seems that BSF addition may lead to bakery products of a high quality, especially if the fermentation process is a long-term one.

### 3.5. Relationships Between Triticale Dough Characteristics with Different Levels of Brewer’s Spent Grain Sourdough Fermentation Addition Incorporated into Its Recipe

The relationships between triticale dough characteristics through principal component analysis (PCA) are shown in Figure 2. The accounted PC1 and PC2 component for PCA is 66.35%, which represents 48.04% and 18.31%, respectively, of the total variance. A closeness between dough development time; stability; and Mixolab parameters C3, C4, C32, C5, and C54 may be noticed which are placed on the left of the graph. Also, a good correlation may be noticed between falling number and C3 and C32 values, a correlation similar to that reported by Codină et al. [45]. According to the PCA graph, the triticale varieties without BSF addition are most closely associated to the dough rheological properties during mixing, pasting and extension. However, higher additions of BSF to triticale flour in dough recipes are closely related to dough rheological properties during fermentation. This behavior may be explained by the increase in the enzymatic activity from triticale dough with a high amount of BSF addition, which improves the gas released from the dough system. Regarding triticale dough samples with their analyzed characteristics, it seems that Ingen 54_10BSG and Ingen 35_10BSG are more related to the dynamic complex modules, and Ingen 54 to the FN, with all of them being the most closely to the Mixolab values related to the starch behavior. These correlations may be due to the high FN of these dough samples, as it is associated with higher *α*-amylase activity. Ingen 35, Ingen 93, and Ingen 33 are more related to the Alveograph and pH values, whereas Ingen 93_10BSG, Ingen 33_10 BSG, and all dough samples with a 17.5% BSF addition are related more to the Rheofermentometer values, Mixolab C12 parameter, and tan δ. These associations are explainable since dough with high amounts of BSF presents a more intense proteolytic and amylolytic activity and becomes more viscous.

The associations between triticale dough samples that were analyzed according to their rheological characteristics are shown in Figure 3. The total variance of the PCA graph was 66.35%, from which PC1 accounts for 48.04%, and PC2 for 18.31%. As may be seen, Ingen 33, Ingen 35, and Ingen 93 triticale varieties are closely associated between them, being placed on the left side of the graph. Generally, the PCs axes clearly distinguish the dough samples without BSF addition and dough samples with BSF addition. It seems that Ingen 93 and Ingen 33 with different amounts of BSF may present a similar technological behavior, being placed on the right side of the PCA graph. Also, a 10% BSF incorporation into the dough recipe may lead to a similar behavior of it for Ingen 54 and Ingen 35 triticale varieties, which are placed on the top left of the PCA graph. A similar behavior may be also noted for Ingen 54 and Ingen 35 with 17.5% BSF addition, as they are placed on the top right of the graph.

## 4. Study’s Limitations

Nowadays, the interest in using triticale for food production has increased due to its valuable nutritional composition and health effects. More due to its high yield potential, resistance to winter, drought and diseases, tolerance to the toxicity of salts, and it is adaptation to the environment, the quantity of triticale cultivated is continuously increasing. However, its possibility to be used to obtain bakery products may be limited due to its technological characteristics. To obtain a high-quality bread, triticale grains must present a certain amount of gluten in order to allow us to obtain a dough with specific elasticity and extensibility. For this purpose, different triticale varieties must be developed with certain characteristics to be used in developing different bakery products. By-products from the brewing industry, like BSG, are of interest since they are not expensive and are available in large amounts. However, its moisture is high after the brewing process, and the possibility of storing it for a long period of time before drying is limited. Even if BSG in a fermented form presents a high nutritional value with less antinutritional compounds its possibility to be used in high amount in bakery products is limited due to the fact that does not contain gluten and may lead to products of low quality. The combination of triticale flour with BSG in a fermented form may lead to food products of high nutritional value, but the technological process must be adjusted to the triticale variety’s technological characteristics, the bakery product type, and the amount of BSF added to the dough recipe.

## 5. Conclusions

Brewer’s spent grain sourdough fermentation (BSF) significantly changes all dough rheological properties, which vary depending on the variety of triticale flour used. The water absorption value decreased with the increased amount of BSF in triticale flour, presenting a possible disadvantage for the bakery producers. Dough stability and dough development time decrease with the increasing level of BSF in dough recipe. Also, a significant decrease could be seen in the protein-weakening value C2 and Mixolab parameters related to triticale flour starch–amylase part (torques C3, C4, and C5) when BSF was incorporated into the dough recipe. The dynamic rheological values show that the BSF addition decreased the G’, G”, and G^*^ modules and increased tan δ values, indicating a decrease in the elasticity structure of the dough samples. For dough rheological properties during extension, BSF addition resulted in a decrease in dough maximum pressure, baking strength, and configuration ratio of the Alveograph curve. The pH values of dough samples decreased with the increase in the level of BSF addition to triticale flour due to the high acidity of BSF and due to the fermentation process from the dough system, leading to an increase in the amounts of acids. During fermentation, all rheological values were improved for dough samples with BSF. We noticed an increase in the total CO_2_ volume production, volume of the gas retained in the dough at the end of the test, maximum height of gaseous production, and retention coefficient. The PCA analysis showed a high association between triticale flour varieties without BSF addition especially for Ingen 35, Ingen 93, and Ingen 33 varieties, thus indicating a similar behavior of it from the rheological point of view. Also, a high association was obtained for dough samples with a 17.5% BSF addition to triticale flour, indicating that it easy for food producers to predict dough behavior when high amounts of BSF are incorporated into dough recipes.

## 6. Future Perspectives

In recent years, increasing attention has been given to the use of triticale and BSG as a by-product that has the potential to generate highly nutritional food products. However, it seems that BSG presents some antinutritional factors, a poor mineral bioavailability and protein digestibility that may be enhanced by different treatments, the most inexpensive one being the fermentation process. Using BSG in a fermented form may be an alternative for food producers in obtaining products of a high quality. Therefore, the possibility of establishing various fermentation processes for BSG may also continue to be explored in order to optimize them from a qualitative point of view so that they can subsequently be used for the development of new food products. Moreover, the possibility of using different BSG types from the brewing industry may lead to new sourdough types which can help us to obtain the best food products. Given the worldwide trend to increase the consumption of bakery products with high nutritional value, with benefits for human health, products that can also be obtained by adding by-products to breadmaking, the possibility of obtaining bread or other food products enriched with different types of BSG obtained through various fermentation process and varieties of triticale grains will continue to be studied.

## Figures and Tables

**Figure 1 foods-14-00041-f001:**
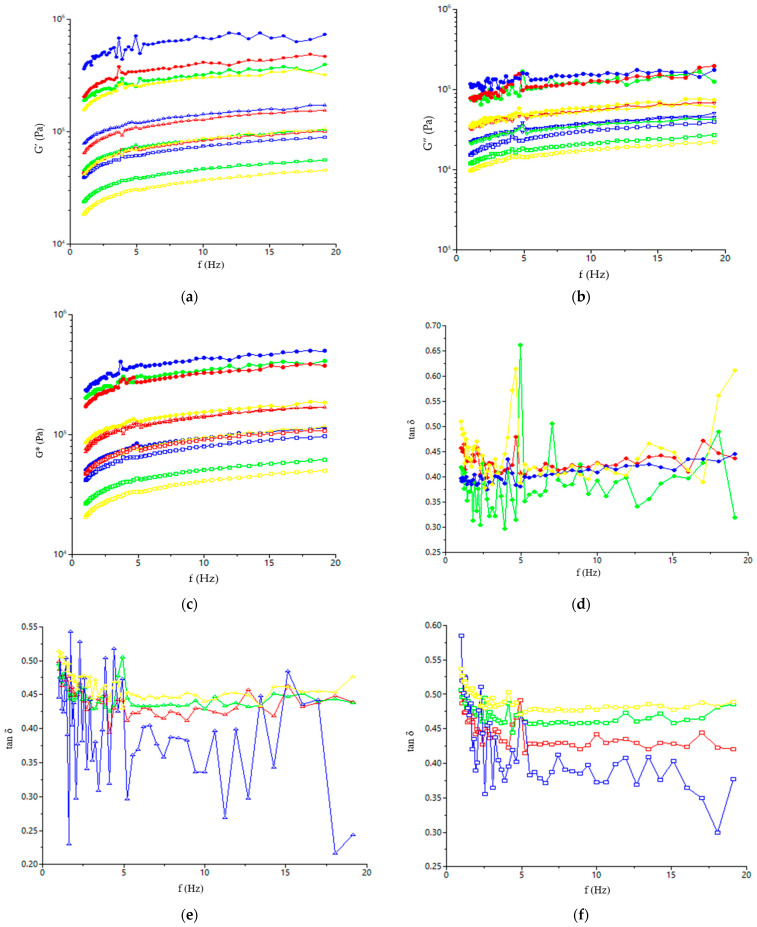
Evaluation with frequency of G’ (**a**), G” (**b**), G* (**c**) and tan *δ* (**d**–**f**) for the dough samples from different triticale varieties and BSF addition (-●- Ingen 33; -●- Ingen 35; -●- Ingen 54; -●- Ingen 93; -∆- Ingen 33_10BSF; -∆- Ingen 35_10BSF; -∆- Ingen 54_10BSF; -∆- Ingen 93_10BSF; -□- Ingen 33_17.5BSF; -□- Ingen 33_17.5BSF; -□- Ingen 33_17.5BSF; -□- Ingen 33_17.5BSF).

**Figure 2 foods-14-00041-f002:**
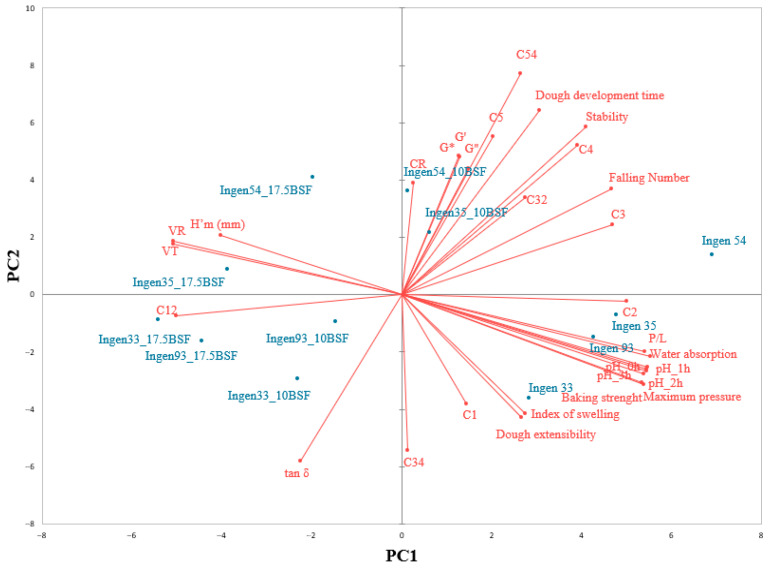
Principal component analysis plot of variables: C1–C5, maximum consistency during stages 1–5, respectively; C12, C32, C34, and C54, difference between Mixolab torques C1 and C2, C3 and C2, C3 and C4, and C5 and C4; H’m, maximum height of gaseous production; VT, total CO_2_ volume production; VR, volume of the gas retained in the dough at the end of the test; CR, retention coefficient.

**Figure 3 foods-14-00041-f003:**
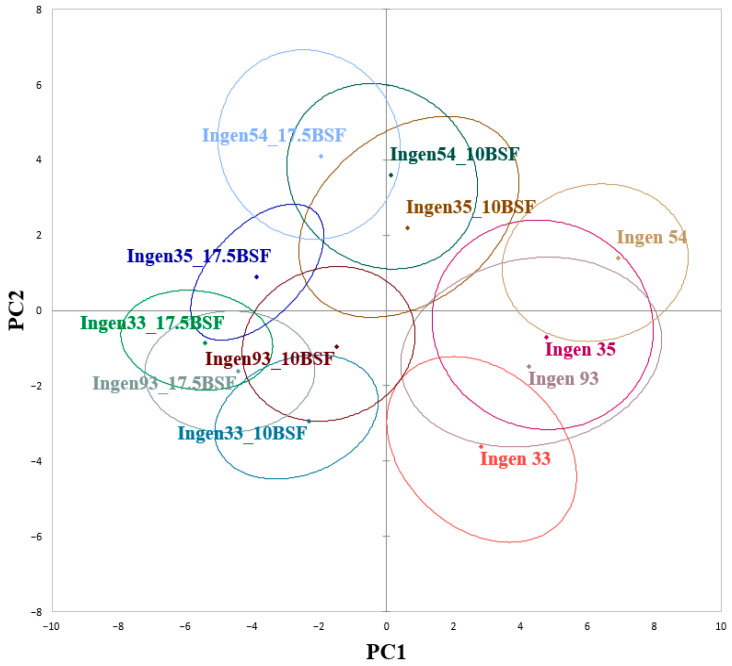
Principal component analysis of dough samples.

**Table 1 foods-14-00041-t001:** Mixolab parameters of triticale flour with different levels of brewer’s spent grain sourdough fermentation.

Samples	WA (%)	ST (min)	DDT (min)	C1 (N∙m)	C2 (N∙m)	C3 (N∙m)	C4 (N∙m)	C5 (N∙m)	C12 (N∙m)	C32 (N∙m)	C34 (N∙m)	C54 (N∙m)
Ingen 33	59.0 ± 0.7 ^g^	2.3 ± 0.05 ^c^	2.53 ± 0.04 ^b^	1.137 ± 0.003 ^g^	0.397 ± 0.002 ^e^	1.618 ± 0.016 ^d^	0.364 ± 0.015 ^c,d^	0.644 ± 0.020 ^c^	0.740 ± 0.007 ^b^	1.221 ± 0.006 ^b^	1.254 ± 0.009 ^f^	0.280 ± 0.007 ^b^
Ingen 33_10BSF	42.0 ± 0.9 ^d,e^	2.2 ± 0.06 ^c^	2.45 ± 0.02 ^a^	1.076 ± 0.002 ^b^	0.204 ± 0.005 ^b^	1.393 ± 0.018 ^b^	0.142 ± 0.009 ^a^	0.327 ± 0.025 ^a^	0.872 ± 0.006 ^e^	1.189 ± 0.008 ^a^	1.251 ± 0.005 ^f^	0.185 ± 0.005 ^a^
Ingen 33_17.5BSF	33.0 ± 0.7 ^a^	1.8 ± 0.08 ^a^	2.58 ± 0.05 ^b^	1.070 ± 0.002 ^a,b^	0.116 ± 0.003 ^a^	1.346 ± 0.014 ^a^	0.119 ± 0.006 ^a^	0.392 ± 0.031 ^a^	0.954 ± 0.008 ^f,g^	1.230 ± 0.009 ^b^	1.227 ± 0.009 ^e,f^	0.273 ± 0.012 ^b^
Ingen 35	59.4 ± 0.9 ^g^	2.9 ± 0.09 ^f^	3.00 ± 0.01 ^c^	1.119 ± 0.004 ^f^	0.365 ± 0.004 ^d,e^	1.782 ± 0.020 ^e^	0.515 ± 0.027 ^e^	0.942 ± 0.018 ^e^	0.754 ± 0.005 ^b^	1.417 ± 0.009 ^e^	1.267 ± 0.012 ^f^	0.427 ± 0.008 ^d^
Ingen 35_10BSF	48.7 ± 0.8 ^f^	2.8 ± 0.04 ^e,f^	3.87 ± 0.06 ^f^	1.120 ± 0.002 ^f^	0.340 ± 0.005 ^d^	1.774 ± 0.017 ^e^	0.786 ± 0.021 ^f^	1.325 ± 0.013 ^g^	0.780 ± 0.007 ^c^	1.434 ± 0.011 ^e,f^	0.988 ± 0.011 ^a^	0.539 ± 0.011 ^f^
Ingen 35_17.5BSF	39.5 ± 0.7 ^c,d^	2.4 ± 0.05 ^d^	2.85 ± 0.0 ^c^	1.113 ± 0.001 ^e^	0.270 ± 0.002 ^c^	1.612 ± 0.021 ^d^	0.491 ± 0.024 ^e^	0.827 ± 0.028 ^d^	0.843 ± 0.009 ^e,d^	1.342 ± 0.006 ^d^	1.121 ± 0.008 ^c,d^	0.336 ± 0.008 ^b,c^
Ingen 54	59.2 ± 0.6 ^g^	3.8 ± 0.07 ^h^	4.08 ± 0.02 ^g^	1.103 ± 0.002 ^d^	0.418 ± 0.006 ^f^	1.977 ± 0.023 ^f^	0.872 ± 0.018 ^g^	1.527 ± 0.032 ^h^	0.685 ± 0.010 ^a^	1.559 ± 0.013 ^g^	1.105 ± 0.010 ^c^	0.655 ± 0.011 ^g,h^
Ingen 54_10BSG	42.0 ± 0.4 ^e^	3.0 ± 0.06 ^f,g^	3.52 ± 0.03 ^e^	1.068 ± 0.002 ^a,b^	0.237 ± 0.004 ^c^	1.546 ± 0.016 ^c^	0.478 ± 0.015 ^e^	0.973 ± 0.027 ^e^	0.831 ± 0.012 ^d,e^	1.309 ± 0.007 ^c^	1.068 ± 0.002 ^b,c^	0.495 ± 0.006 ^e^
Ingen 54_17.5BSF	37.0 ± 0.8 ^c^	3.1 ± 0.08 ^g^	3.03 ± 0.01 ^c,d^	1.085 ± 0.001 ^c^	0.209 ± 0.005 ^b^	1.612 ± 0.023 ^d^	0.454 ± 0.010 ^d,e^	1.104 ± 0.024 ^f^	0.876 ± 0.002 ^e^	1.403 ± 0.012 ^e^	1.158 ± 0.007 ^d^	0.650 ± 0.009 ^g,h^
Ingen 93	58.3 ± 0.2 ^g^	2.6 ± 0.05 ^d,e^	2.85 ± 0.0 ^c^	1.113 ± 0.001 ^e^	0.344 ± 0.006 ^d^	1.765 ± 0.019 ^e^	0.450 ± 0.014 ^d,e^	0.817 ± 0.031 ^d^	0.769 ± 0.008 ^b,c^	1.421 ± 0.009 ^e,f^	1.315 ± 0.009 ^g^	0.367 ± 0.012 ^c,d^
Ingen 93_10BSF	42.0 ± 0.5 ^e^	2.6 ± 0.04 ^d,e^	3.07 ± 0.02 ^d^	1.122 ± 0.001 ^f^	0.134 ± 0.002 ^a^	1.781 ± 0.016 ^e^	0.285 ± 0.012 ^c^	0.701 ± 0.024 ^c^	0.988 ± 0.012 ^g,h^	1.647 ± 0.002 ^h^	1.496 ± 0.004 ^h^	0.416 ± 0.008 ^d^
Ingen 93_17.5BSF	35.0 ± 0.8 ^b^	2.0 ± 0.6 ^b^	2.45 ± 0.01 ^a^	1.139 ± 0.002 ^g^	0.194 ± 0.004 ^b^	1.365 ± 0.006 ^a^	0.228 ± 0.024 ^b^	0.494 ± 0.021 ^b^	0.945 ± 0.009 ^f,g^	1.171 ± 0.005 ^a^	1.137 ± 0.005 ^d^	0.266 ± 0.005 ^b^

Mixolab parameters: ST—stability; DDT—dough development time; C1–C5—maximum consistency during stages 1–5, respectively; C12—difference between torques C1 and C2; C32—difference between torques C3 and C2; C34—difference between torques C3 and C4; C54—difference between torques C5 and C4; Samples: Ingen 33, Ingen 35, Ingen 54, and Ingen 93—triticale flour varieties; Ingen 33_10BSF, Ingen 35_10BSF, Ingen 54_10BSF, and Ingen 93_10BSF—triticale flour varieties with 10% brewer’s spent grain sourdough fermentation addition; Ingen 33_17.5BSF, Ingen 35_17.5BSF, Ingen 54_17.5BSF, and Ingen 93_17.5 BSF—triticale flour varieties with 17.5% brewer’s spent grain sourdough fermentation addition; ^a–h^ Means in the same column with different letters for each sample type are significantly different (*p* < 0.05).

**Table 2 foods-14-00041-t002:** Elastic modulus (G’), loss modulus (G”), complex modulus (G*), and tan δ evolution of the dough samples with triticale flour with different levels of brewer’s spent grain sourdough fermentation addition.

Dough Samples	G’ at 10 Hz (Pa)	G” at 10 Hz (Pa)	G* at 10 Hz (Pa)	tan δ at 10 Hz
Ingen 33	142,400 ± 135 ^b^	61,010 ± 98 ^c^	154,900 ± 119 ^c^	0.428 ± 0.0013 ^e^
Ingen 33_10BSF	128,000 ± 119 ^b^	54,940 ± 103 ^c^	139,300 ± 106 ^b^	0.429 ± 0.0007 ^e^
Ingen 33_17.5BSF	83,050 ± 106 ^b^	36,670 ± 58 ^b^	90,790 ± 78 ^b^	0.441 ± 0.0011 ^f^
Ingen 35	319,600 ± 98 ^c^	125,500 ± 127 ^d^	343,300 ± 101 ^d^	0.392 ± 0.0025 ^c^
Ingen 35_10BSF	86,650 ± 103 ^b^	37,150 ± 89 ^b^	94,280 ± 173 ^b^	0.428 ± 0.0016 ^e^
Ingen 35_17.5BSF	46,290 ± 120 ^a^	21,280 ± 78 ^a^	50,950 ± 115 ^a^	0.459 ± 0.0027 ^g^
Ingen 54	141,900 ± 109 ^b^	63,200 ± 113 ^b^	258,500 ± 98 ^b^	0.445 ± 0.0019 ^d^
Ingen 54_10BSF	123,400 ± 201 ^e^	58,400± 106 ^f^	158,000 ± 131 ^f^	0.473 ± 0.0003 ^a^
Ingen 54_17.5BSF	89,800 ± 114 ^d^	42,700 ± 27 ^e^	97,400 ± 96 ^e^	0.475 ± 0.0009 ^b^
Ingen 93	301,200 ± 127 ^c^	128,500 ± 76 ^d^	327,500 ± 125 ^d^	0.426 ± 0.0015 ^e^
Ingen 93_10BSF	87,690 ± 128 ^b^	38,580 ± 49 ^b^	95,800 ± 76 ^b^	0.439 ± 0.0021 ^f^
Ingen 93_17.5BSF	37,000 ± 101 ^a^	17,690 ± 38 ^a^	41,010 ± 81 ^a^	0.478 ± 0.0018 ^h^

G’—elastic modulus; G”—loss modulus; G*—complex modulus; tan δ—loss tangent. Samples: Ingen 33, Ingen 35, Ingen 54, Ingen 93—triticale flour varieties; Ingen 33_10BSF, Ingen 35_10BSF, Ingen 54_10BSF, Ingen 93_10 BSF—triticale flour varieties with 10% brewer’s spent grain sourdough fermentation addition; Ingen 33_17.5BSF, Ingen 35_17.5BSF, Ingen 54_17.5BSF, Ingen 93_17.5 BSF—triticale flour varieties with 17.5% brewer’s spent grain sourdough fermentation addition. ^a–h^—mean values in the same column followed by different letters are significantly different (*p* < 0.05).

**Table 3 foods-14-00041-t003:** Alveograph data of dough with different levels of brewer’s spent grain sourdough fermentation addition to triticale flour.

Dough Samples	P (mm)	L (mm)	G	W (10^−4^ J)	P/L
Ingen 33	100 ± 1.5 ^e^	37 ± 0.6 ^g^	13.5 ± 0.01 ^g^	114 ± 2 ^e^	2.70 ± 0.10 ^d^
Ingen 33_10BSF	53 ± 1.0 ^c^	39 ± 0.8 ^h^	13.9 ± 0.04 ^g^	51 ± 2 ^c^	1.35 ± 0.09 ^b^
Ingen 33_17.5BSF	32 ± 0.7 ^a^	28 ± 0.5 ^d^	11.8 ± 0.03 ^d^	33 ± 1 ^b^	1.14 ± 0.02 ^a^
Ingen 35	102 ± 2.0 ^e^	31 ± 0.7 ^e,f^	12.4 ± 0.06 ^d^	112 ± 2 ^e^	3.29 ± 0.05 ^f^
Ingen 35_10BSF	50 ± 1.6 ^c^	24 ± 0.4 ^b^	10.9 ± 0.03 ^b,c^	47 ± 1 ^c^	2.08 ± 0.06 ^c^
Ingen 35_17.5BSF	27 ± 1.3 ^a^	19 ± 0.1 ^a^	9.7 ± 0.01 ^a^	18 ± 1 ^a^	1.42 ± 0.02 ^b^
Ingen 54	106 ± 1.8 ^e^	32 ± 0.5 ^f^	12.6 ± 0.05 ^e,f^	129 ± 4 ^f^	3.31 ± 0.03 ^f^
Ingen 54_10BSF	43 ± 0.6 ^b^	33 ± 0.2 ^f^	12.8 ± 0.04 ^e,f^	45 ± 2 ^c^	1.30 ± 0.06 ^b^
Ingen 54_17.5BSF	26 ± 1.0 ^a^	25 ± 0.4 ^c^	11.1 ± 0.01 ^c^	20 ± 1 ^a^	1.04 ± 0.02 ^a^
Ingen 93	95 ± 1.7 ^d^	32 ± 0.6 ^f^	12.6 ± 0.03 ^e,f^	99 ± 2 ^d^	2.97 ± 0.03 ^e^
Ingen 93_10BSF	41 ± 1.1 ^b^	30 ± 0.5 ^e^	12.2 ± 0.04 ^e^	47 ± 2 ^c^	1.37 ± 0.02 ^b^
Ingen 93_17.5BSF	25 ± 0.8 ^a^	23 ± 0.3 ^a^	10.7 ± 0.02 ^b^	23 ± 1 ^a^	1.08 ± 0.01 ^a^

P, tenacity (maximum pressure); L, dough extensibility; G, index of swelling; W, baking strength; P/L, configuration ratio of the Alveograph curve. Samples: Ingen 33, Ingen 35, Ingen 54, Ingen 93—triticale flour varieties; Ingen 33_10BSF, Ingen 35_10BSF, Ingen 54_10BSF, Ingen 93_10 BSF—triticale flour varieties with 10% brewer’s spent grain sourdough fermentation addition; Ingen 33_17.5BSF, Ingen 35_17.5BSF, Ingen 54_17.5BSF, Ingen 93_17.5 BSF—triticale flour varieties with 17.5% brewer’s spent grain sourdough fermentation addition. ^a–h^—mean values in the same column followed by different letters are significantly different (*p* < 0.05).

**Table 4 foods-14-00041-t004:** pH values of dough samples with different levels of brewer’s spent grain sourdough fermentation addition to triticale flour.

Samples	pH			
	0 h	1 h	2 h	3 h
Ingen 33	6.15 ± 0.02 ^j,k^	6.04 ± 0.03 ^h,i^	5.98 ± 0.01 ^g,h^	5.95 ± 0.02 ^g^
Ingen 33_10BSF	5.74 ± 0.04 ^e,f^	5.71 ± 0.02 ^d,e^	5.67 ± 0.03 ^d,e^	5.64 ± 0.03 ^c,d^
Ingen 33_17.5BSF	5.64 ± 0.03 ^c,d^	5.60 ± 0.03 ^b,c^	5.55 ± 0.04 ^b,c^	5.51 ± 0.04 ^a,b^
Ingen 35	6.10 ± 0.02 ^i,j^	6.02 ± 0.01 ^h^	5.97 ± 0.01 ^g^	5.94 ± 0.01 ^g^
Ingen 35_10BSF	5.70 ± 0.01 ^d,e^	5.68 ± 0.03 ^d,e^	5.62 ± 0.02 ^c,d^	5.59 ± 0.03 ^b,c^
Ingen 35_17.5BSF	5.58 ± 0.02 ^b,c^	5.55 ± 0.02 ^b^	5.52± 0.04 ^a,b^	5.48 ± 0.02 ^a^
Ingen 54	6.18 ± 0.03 ^j,k^	6.08 ± 0.02 ^i^	6.04 ± 0.01 ^h,i^	5.97 ± 0.01 ^g^
Ingen 54_10BSF	5.76 ± 0.01 ^e,f^	5.73 ± 0.03 ^e,f^	5.70 ± 0.02 ^d,e^	5.67 ± 0.03 ^d,e^
Ingen 54_17.5BSF	5.66 ± 0.02 ^d^	5.63 ± 0.04 ^c,d^	5.58 ± 0.02 ^b,c^	5.54 ± 0.03 ^a,b^
Ingen 93	6.12 ± 0.02 ^i,j^	6.00 ± 0.01 ^h^	5.97 ± 0.01 ^g^	5.94 ± 0.01 ^g^
Ingen 93_10BSF	5.72 ± 0.01 ^e^	5.70 ± 0.03 ^d,e^	5.65 ± 0.03 ^c,d^	5.62 ± 0.02 ^c,d^
Ingen 93_17.5BSF	5.61 ± 0.03 ^d,e^	5.58 ± 0.04 ^b,c^	5.54 ± 0.02 ^b^	5.49 ± 0.02 ^a^

Samples: Ingen 33, Ingen 35, Ingen 54, and Ingen 93—triticale flour varieties; Ingen 33_10BSF, Ingen 35_10BSF, Ingen 54_10BSF, and Ingen 93_10 BSF—triticale flour varieties with 10% brewer’s spent grain sourdough fermentation addition; Ingen 33_17.5BSF, Ingen 35_17.5BSF, Ingen 54_17.5BSF, and Ingen 93_17.5 BSF—triticale flour varieties with 17.5% brewer’s spent grain sourdough fermentation addition. ^a–k^ Mean values in the same column followed by different letters are significantly different (*p* < 0.05).

**Table 5 foods-14-00041-t005:** Rheofermentometer and falling-number data of triticale dough with different levels of brewer’s spent grain sourdough fermentation addition.

Dough Samples	H’m (mm)	VT (mL)	VR (mL)	CR (%)	FN (s)
Ingen 33	87.1 ± 0.7 ^a,b^	1647 ± 17 ^a^	1640 ± 5 ^a^	99.5 ± 0.02 ^f^	73 ± 1 ^b^
Ingen 33_10BSF	87.9 ± 0.5 ^b^	1879 ± 21 ^c^	1864 ± 0 ^c^	99.2 ± 0.04 ^c^	68 ± 2 ^a,b^
Ingen 33_17.5BSF	98.4 ± 0.1 ^f^	2120 ± 10 ^g^	2109 ± 3 ^g^	99.5 ± 0.01 ^d,e^	64 ± 2 ^a^
Ingen 35	88.7 ± 0.3 ^b,c^	1921 ± 15 ^d^	1907 ± 4 ^c^	99.2 ± 0.02 ^c^	102 ± 1 ^e^
Ingen 35_10BSF	89.2 ± 0.2 ^c^	2042 ± 18 ^e,f^	2013 ± 3 ^e^	98.6 ± 0.01 ^a^	77 ± 2 ^c^
Ingen 35_17.5BSF	100.0 ± 0.9 ^g^	2213 ± 15 ^h^	2198 ± 8 ^h^	99.3 ± 0.03 ^d^	68 ± 1 ^a,b^
Ingen 54	87.4 ± 0.3 ^a,b^	1631 ± 12 ^a^	1620 ± 5 ^a^	99.3 ± 0.01 ^d^	136 ± 2 ^h^
Ingen 54_10BSF	87.8 ± 0.3 ^b^	1948 ± 15 ^d^	1933 ± 3 ^c^	99.2 ± 0.2 ^c^	111 ± 1 ^f^
Ingen 54_17.5BSF	89.1 ± 0.2 ^c^	1983 ± 13 ^e^	1968 ± 8 ^d^	99.2 ± 0.01 ^c^	93 ± 1 ^d^
Ingen 93	87.2 ± 0.1 ^a^	1743 ± 2 ^b^	1727 ± 5 ^b^	99.1 ± 0.02 ^b^	120 ± 1 ^g^
Ingen 93_10BSF	90.8 ± 0.5 ^d^	2024 ± 9 ^e^	2009 ± 4 ^d,e^	99.3 ± 0.03 ^d^	70 ± 2 ^b^
Ingen 93_17.5BSF	94.0 ± 0.1 ^e^	2128 ± 11 ^g^	2115 ± 3 ^g^	99.4 ± 0.02 ^e^	64 ± 1 ^a^

H’m—maximum height of gaseous production; VT—total CO_2_ volume production; VR—volume of the gas retained in the dough at the end of the test; CR—retention coefficient. Samples: Ingen 33, Ingen 35, Ingen 54, and Ingen 93—triticale flour varieties; Ingen 33_10BSF, Ingen 35_10BSF, Ingen 54_10BSF, and Ingen 93_10 BSF—triticale flour varieties with 10% brewer’s spent grain sourdough fermentation addition; Ingen 33_17.5BSF, Ingen 35_17.5BSF, Ingen 54_17.5BSF, and Ingen 93_17.5 BSF—triticale flour varieties with 17.5% brewer’s spent grain sourdough fermentation addition. ^a–h^ Mean values in the same column followed by different letters are significantly different (*p* < 0.05).

## Data Availability

The original contributions presented in this study are included in the article. Further inquiries can be directed to the corresponding author.

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
