# Peer review of "Effect of Brewers’ Spent Grain Addition to a Fermented Form on Dough Rheological Properties from Different Triticale Flour Cultivars"

_foods, 2024, doi:10.3390/foods14010041_

Round 1
Reviewer 1 Report
Comments and Suggestions for Authors
In the presented manuscript, the authors studied the rheological properties of triticale dough with the addition of brewers' spent grain. However, I do not see the value of this manuscript. The authors added an arbitrary amount of brewers' spent grain, which resulted in the deterioration of triticale flour quality. As a consequence, the quality parameters obtained for the enriched dough disqualified it for baking purposes.
Please include the optimal values of individual parameters, such as water absorption, development time, etc., for triticale flour, and demonstrate how the additive affects the quality of triticale flour.
The brewers' spent grain sourdough was incorporated into the triticale dough at levels of 10% and 17.5%. What was the rationale for incorporating such amounts into the triticale flour?
Line 41-42: "On average, triticale flour is characterized by a relatively high content of starch (>70 g/100 g), protein (>13 g/100 g), and dietary fiber (>14 g/100 g)." Please provide the range or average values of these individual components in parentheses, rather than using the ">" symbol.
Line 52-53: "Blending triticale flour with other types of flour can balance these shortcomings, providing rheological stability and improved bakery products." Please specify which products are improved and include a citation to support this claim.
Line 224: "Significance level of p < 0.05" should be "Significance level of α = 0.05." The p-value is calculated, not assumed.
The authors did not properly compare their results with those of other authors who worked on the same cultivars (https://doi.org/10.3390/foods13111671).
Author Response
23 December 2024
Dear Referee,
We would like to thank the referee for the close reading and for the proper suggestions. We hope that we provide all the answers to the reviewer’s comments.
Thank you very much for the recommendations to publish our paper entitled “Effect of brewers’ spent grain addition on dough rheological properties from different triticale flours cultivars”.
The present version of the paper has been revised according to the reviewer’s suggestions.
We uploaded the corrected version of the article for which we used the red/blue colour for the addition text.
GENERAL COMMENTS:
Referee comments: In the presented manuscript, the authors studied the rheological properties of triticale dough with the addition of brewers' spent grain. However, I do not see the value of this manuscript. The authors added an arbitrary amount of brewers' spent grain, which resulted in the deterioration of triticale flour quality. As a consequence, the quality parameters obtained for the enriched dough disqualified it for baking purposes.
Response: We would like the referee for the close reading of our manuscript. The addition of brewers' spent grain in a fermented form (BSF) in this amount has been chosen in order to obtain in the future bakery products and pasta of an improved nutritional value ( low levels of BSF will not help us a lot to obtain good results from a nutritional perspective). We completed now in the manuscript these argues. Also a previous study made by other researchers (https://www.sciencedirect.com/science/article/abs/pii/S2213329121000496) on pasta nutritional characteristics with BSF lead us to the conclusion that we must use at least 15% BSF in order to have good nutritional results for bread and pasta in the future. We have obtained (data not shown in the manuscript) pasta of a very good quality with this level of BSF addition. Also, we have been obtained bread of a good quality in our previous study (https://www.mdpi.com/2304-8158/13/11/1671) only from triticale flour which encourage us to use such high levels of BSF addition. Indeed, we obtained bad results to dough rheological properties during extension (alveograph) but we think that this is due to the standard method used for this device (we explained in the text). To the rest of the rheological data we think that the results obtained are not so bad. To Mixolab, for some triticale varieties at 10% BSF addition dough stability did not presented significant changes. Also, for Ingen 54 and also Ingen 33 we think that we can obtain bakery products with high quality for 17.5% BSF addition according to our data obtained. The bakery products seem to have a lower retrogradation which may be an advantage. More, the data obtained to Rheofermentometer are impressive. The dough rheological behavior during fermentation has been improved and we concluded that a long fermentation time will help us to obtain bread of a high quality compared to the control sample with both addition levels of BSF (10 and 17.5%).
Referee comments: Please include the optimal values of individual parameters, such as water absorption, development time, etc., for triticale flour, and demonstrate how the additive affects the quality of triticale flour.
Response: We would like to thank to the referee for his/her suggestions. We completed the manuscript with more informations related to the values obatined and it impact on the quality of triticale flour.
Referee comments: The brewers' spent grain sourdough was incorporated into the triticale dough at levels of 10% and 17.5%. What was the rationale for incorporating such amounts into the triticale flour?
Response: We response previously. The addition of brewers' spent grain in a fermented form (BSF) in this amount has been chosen in order to obtain in the future bakery products and pasta of an improved nutritional value (low levels of BSF will not help us a lot to obtain good results from a nutritional perspective). We completed now in the manuscript these argues. Also a previous study made by other researchers (https://www.sciencedirect.com/science/article/abs/pii/S2213329121000496) on pasta nutritional characteristics with BSF lead us to the conclusion that we must use at least 15% BSF in order to have good nutritional results for bread and pasta in the future. We have obtained (data not shown in the manuscript) pasta of a very good quality with this level of BSF addition. Also, we have been obtained bread of a good quality in our previous study (https://www.mdpi.com/2304-8158/13/11/1671) only from triticale flour which encourage us to use such high levels of BSF addition. According to the rheological data in especially during fermentation we really think that we will also obtain bread of a good quality. We expect the best results to Ingen 54 and Ingen 33 varieties (according to Mixolab results).
Referee comments: Line 41-42: "On average, triticale flour is characterized by a relatively high content of starch (>70 g/100 g), protein (>13 g/100 g), and dietary fiber (>14 g/100 g)." Please provide the range or average values of these individual components in parentheses, rather than using the ">" symbol.
Response: We have provided the range or average values of these individual components in parentheses, rather than using the ">" symbol according to the referee suggestions.
Referee comments: Line 52-53: "Blending triticale flour with other types of flour can balance these shortcomings, providing rheological stability and improved bakery products." Please specify which products are improved and include a citation to support this claim
Response: We have now completed and justify in the manuscript according to the referee suggestions.
Referee comments: Line 224: "Significance level of p < 0.05" should be "Significance level of α = 0.05." The p-value is calculated, not assumed.
Response: We want to thank to the referee for the close reading of our manuscript. We changed according to the referee suggestions.
Referee comments: The authors did not properly compare their results with those of other authors who worked on the same cultivars (https://doi.org/10.3390/foods13111671).
Response: We want to thank to the referee for the close reading of our manuscript. This was our previously published article. The data of triticale varieties presented some slightly change due to the flour maturation (has been 5 months between the determinations). However, the objective of this study was to analyze the BSF effect on dough rheological behavior.
Finnaly, we want to thank the reviewer for all the comments and his/her recommendations. We really believe that help us to improve the quality of our manuscript.
Sincerely,
Georgiana Codina et al.
Reviewer 2 Report
Comments and Suggestions for Authors
Review on manuscript: foods-3373675
Effect of brewers’ spent grain addition on dough rheological properties from different triticale flours cultivars
by Aliona Ghendov-Mosanu, Sorina Ropciuc, Adriana Dabija, Olesea Saitan, Olga Boestean, Sergiu Paiu, Iurie Rumeus Svetlana Leatamborg, Galina Lupascu, and Georgiana Gabriela Codină
submitted to Foods
In the manuscript submitted for review, the authors studied the effect of brewers’ spent grain on rheological properties of triticale dough. In my opinion, the manuscript is generally prepared correctly, so after some revision could be accepted for publication.
Detailed recommendation:
Abstract – the most important results in numerical form should be added,
lines 17-18 – Dynamic Rheometer is an incorrect term,
Introduction – could be shortened,
line 102 – Italic style should be used,
line 125 – incorrect reference citation,
lines 156-157 – where exactly did the research material come from?
line 161 – missing units,
line 168 – what exactly does room temperature mean?
line 235 – should be: stability time (ST),
line 272 – the gelatinization process concerns starch and not dough,
line 275 – should be: starch granules will gelatinize; gelation is a different process,
line 299 – should be: gelatinized starch,
Table 2 – tan δ values ​​should be given to the third significant digit,
Figure 1 – should be divided and marked with letters, and the legend should be modified accordingly, the last graph is difficult to read,
line 608 – incomplete title of the journal.
Comments on the Quality of English Language
see report
Author Response
23 December 2024
Dear Referee,
We would like to thank the referee for the close reading and for the proper suggestions. We hope that we provide all the answers to the reviewer’s comments.
Thank you very much for the recommendations to publish our paper entitled “Effect of brewers’ spent grain addition on dough rheological properties from different triticale flours cultivars”.
The present version of the paper has been revised according to the reviewer’s suggestions.
We uploaded the corrected version of the article for which we used the red/blue colour for the addition text.
GENERAL COMMENTS:
Referee comments: In the manuscript submitted for review, the authors studied the effect of brewers’ spent grain on rheological properties of triticale dough. In my opinion, the manuscript is generally prepared correctly, so after some revision could be accepted for publication.
Response: We would like the referee for the close reading of our manuscript and his/her appreciation.
Referee comments: Abstract – the most important results in numerical form should be added,
Response: We would like to thank to the referee for his/her suggestions. We completed the abstract with more numerical data. However, due to the limitations lenght of the abstract we could not provide numeral data for all our rheological values.
Referee comments: lines 17-18 – Dynamic Rheometer is an incorrect term,
Response: We would like to thank the referee for his/her suggestions. We changed.
Referee comments: Introduction – could be shortened,
Response: We would like to thank the referee for his/her suggestions. We shortened it according to the referee suggestions.
Referee comments: line 102 – Italic style should be used,
Response: We would like to thank the referee for his/her suggestions. We modified.
Referee comments: line 125 – incorrect reference citation,
Response: We would like to thank the referee for his/her suggestions. We modified.
Referee comments: lines 156-157 – where exactly did the research material come from?
Response: We completed the manuscript with these information’s.
Referee comments: line 161 – missing units,
Response: We completed.
Referee comments: line 168 – what exactly does room temperature mean?
Response: We completed in the manuscript.
Referee comments: line 235 – should be: stability time (ST),
Response: We revised.
Referee comments: line 272 – the gelatinization process concerns starch and not dough,
Response: We would like to thank the referee for his/her suggestions. We revised.
Referee comments: line 275 – should be: starch granules will gelatinize; gelation is a different process,
Response: We would like to thank the referee for his/her suggestions. We revised.
Referee comments: line 299 – should be: gelatinized starch,
Response: We would like to thank the referee for his/her suggestions. We revised.
Referee comments: Table 2 – tan δ values should be given to the third significant digit,
Response: We would like to thank the referee for his/her suggestions. We revised.
Referee comments: Figure 1 – should be divided and marked with letters, and the legend should be modified accordingly, the last graph is difficult to read,
Response: We would like to thank the referee for his/her suggestions. We divided the last graph according to the referee’s suggestions.
Referee comments: line 608 – incomplete title of the journal.
Response: We completed according to the referee suggestions.
Finnaly, we want to thank the reviewer for all the comments and his/her recommendations. We really believe that help us to improve the quality of our manuscript.
Sincerely,
Georgiana Codina et al.
Reviewer 3 Report
Comments and Suggestions for Authors
Dear authors,
Please, conduct the following corrections before manuscript publication:
L160 - Please, cite the method (inserting references)
L161- include the '%' for protein and ash content. Please, specify from where the gluten results was expressed (did the 19.01-27.21% come from the 13.19-15.1 protein content?)
L165 - Include information from where the BSG was taken (beer facility?). Also, write the company, city, country from where BSG was taken.
L166 - 'Humidity' express water content in the air. The term 'moisture' express the amount of water in materials. Also, write how moisture content was measured to confirm the value of 6.3% and cite the method.
L174 - The pH and acidity WERE determined (not 'was').
L176 - Cite the method.
L180/L205/L213 - 'have' or has?
Table 3 - Pressure (P) is defined by Force x Area (basic unit: N/m2). Double check the unit for extensibility (L) and the P/L ratio. Also, the index of swelling was supposed to be dimensionless
L383 - HAS
Author Response
23 December 2024
Dear Referee,
We would like to thank the referee for the close reading and for the proper suggestions. We hope that we provide all the answers to the reviewer’s comments.
Thank you very much for the recommendations to publish our paper entitled “Effect of brewers’ spent grain addition on dough rheological properties from different triticale flours cultivars”.
The present version of the paper has been revised according to the reviewer’s suggestions.
We uploaded the corrected version of the article for which we used the red/blue colour for the addition text.
GENERAL COMMENTS:
Referee comments: L160 - Please, cite the method (inserting references)
Response: We would like to thank to the referee for his/her close reading of our manuscript and suggestions. We completed the abstract with method citetion.
Referee comments: L161- include the '%' for protein and ash content. Please, specify from where the gluten results was expressed (did the 19.01-27.21% come from the 13.19-15.1 protein content?)
Response: We would like to thank to the referee for his/her suggestions. We completed the manuscript with the required informations according to the referee suggestions.
Referee comments: L165 - Include information from where the BSG was taken (beer facility?). Also, write the company, city, country from where BSG was taken.
Response: We would like to thank to the referee for his/her suggestions. We included in the manuscript the required informations according to the referee suggestions.
Referee comments: L166 - 'Humidity' express water content in the air. The term 'moisture' express the amount of water in materials. Also, write how moisture content was measured to confirm the value of 6.3% and cite the method.
Response: We would like to thank to the referee for his/her remarks. We revised and we completed in the manucript the method used and we cited according to the referee suggestions.
Referee comments: L174 - The pH and acidity WERE determined (not 'was').
Response: We revised.
Referee comments: L176 - Cite the method.
Response: We cited.
Referee comments: L180/L205/L213 - 'have' or has?
Response: We revised. Correct is has. Thank you so much for your close reading of our manuscript!
Referee comments: Table 3 - Pressure (P) is defined by Force x Area (basic unit: N/m2). R: tenacity (maximum pressure required for the deformation of the sample) Double check the unit for extensibility (L) and the dau P/L ratio. Also, the index of swelling was supposed to be dimensionless
Response: We cheked. According to the standard method the unit is correct. Pressure is also defined as dough tenacity. The index of swelling is dimensionless. Thank you so much for your remark and we revised.
Referee comments: L383 – HAS
Response: We revised. Thank you so much for your close reading of our manuscript!
Finnaly, we want to thank the reviewer for all the comments and his/her recommendations. We really believe that help us to improve the quality of our manuscript.
Sincerely,
Georgiana Codina et al.
Reviewer 4 Report
Comments and Suggestions for Authors
This is a very pertinent study that deserves to be considered for publication in Foods. However, I have some suggestions for the authors.
In the abstract, before the study’s aim, you should include a background statement. The main results (values with statistical significance) should be mentioned, as well as practical implications and conclusions based on the obtained results.
At the end of the introductory section, the study’s objectives need to be clarified.
The inclusion of a figure with an illustration of the steps you made to conduct your study would be recommended in the Materials and Methods section.
Can you please provide more data from other studies carried out worldwide similar to yours and discuss the results with yours? This would bring an international impact to your work.
I suggest you include a new section with the study’s limitations.
What can be done next? Elaborate on future perspectives after the Conclusions.
Author Response
23 December 2024
Dear Referee,
We would like to thank the referee for the close reading and for the proper suggestions. We hope that we provide all the answers to the reviewer’s comments.
Thank you very much for the recommendations to publish our paper entitled “Effect of brewers’ spent grain addition on dough rheological properties from different triticale flours cultivars”.
The present version of the paper has been revised according to the reviewer’s suggestions.
We uploaded the corrected version of the article for which we used the red/blue colour for the addition text.
GENERAL COMMENTS:
Referee comments: This is a very pertinent study that deserves to be considered for publication in Foods. However, I have some suggestions for the authors.
Response: We would like to thank to the referee for his/her appreciations.
Referee comments: In the abstract, before the study’s aim, you should include a background statement. The main results (values with statistical significance) should be mentioned, as well as practical implications and conclusions based on the obtained results.
Response: We revised our manuscript according to the referee suggestions.
Referee comments: At the end of the introductory section, the study’s objectives need to be clarified.
Response: We completed the end of the introduction with the sudy objectives according to referee suggestion.
Referee comments: The inclusion of a figure with an illustration of the steps you made to conduct your study would be recommended in the Materials and Methods section.
Response: We made a graphical abstract which illustrate our steps to conduct our study.
Referee comments: Can you please provide more data from other studies carried out worldwide similar to yours and discuss the results with yours? This would bring an international impact to your work.
Response: We tried to complete our manuscript with similar data. Unfortnally only 2 studies has been made until now on BSF addition in samolina and wheat flour, so we do not have other studies to compare our data. No study has been made on BSF addition in trticale flour.
Referee comments: I suggest you include a new section with the study’s limitations.
Response: We completed our manuscript with a new section with the study’s limitations.
Referee comments: What can be done next? Elaborate on future perspectives after the Conclusions.
Response: We completed our manuscript with future prespectives according to the referee suggestions.
Finnaly, we want to thank the reviewer for all the comments and his/her recommendations. We really believe that help us to improve the quality of our manuscript.
Sincerely,
Georgiana Codina et al.
Round 2
Reviewer 1 Report
Comments and Suggestions for Authors
The authors corrected the manuscript accordingly.